# An Alkaloid from a Highly Invasive Seaweed Increases the Voracity and Reproductive Output of a Model Fish Species

**DOI:** 10.3390/md20080513

**Published:** 2022-08-12

**Authors:** Valentina Schiano, Adele Cutignano, Daniela Maiello, Marianna Carbone, Maria Letizia Ciavatta, Gianluca Polese, Federica Fioretto, Chiara Attanasio, Antonio Palladino, Serena Felline, Antonio Terlizzi, Livia D’Angelo, Paolo de Girolamo, Mimmo Turano, Carla Lucini, Ernesto Mollo

**Affiliations:** 1Institute of Biomolecular Chemistry, National Research Council of Italy, 80078 Pozzuoli, Italy; 2Department of Veterinary Medicine and Animal Productions, University of Naples “Federico II”, 80131 Naples, Italy; 3Department of Biology, University of Naples “Federico II”, 80126 Naples, Italy; 4School of Biosciences and Veterinary Medicine, University of Camerino, 62023 Camerino, Italy; 5Department of Agricultural Science, University of Naples “Federico II”, 80055 Portici, Italy; 6The National Interuniversity Consortium for Marine Sciences (CoNISMa), 00198 Rome, Italy; 7Stazione Zoologica Anton Dohrn, 80121 Naples, Italy; 8Department of Life sciences, University of Trieste, 34128 Trieste, Italy

**Keywords:** marine biological invasions, *Caulerpa*, caulerpin, zebrafish, food intake, reproduction

## Abstract

The invasive macroalga *Caulerpa cylindracea* has spread widely in the Mediterranean Sea, becoming a favorite food item for native fish for reasons yet unknown. By using a combination of behavioral, morphological, and molecular approaches, herein we provide evidence that the bisindole alkaloid caulerpin, a major secondary metabolite of *C. cylindracea*, significantly increases food intake in the model fish *Danio rerio*, influencing the regulation of genes involved in the orexigenic pathway. In addition, we found that the compound improves fish reproductive performance by affecting the hypothalamus–pituitary–gonadal axis. The obtained results pave the way for the possible valorization of *C. cylindracea* as a sustainable source of a functional feed additive of interest to face critical challenges both in aquaculture and in human nutrition.

## 1. Introduction

Among marine invasive species posing serious ecological and economic threats to the Mediterranean, the green alga *Caulerpa cylindracea* has become a major component of the diet of native fish [1,2]. *C. cylindracea* belongs to a group of algae collectively called “sea grapes” that are already exploited for human consumption [3,4]. Its high palatability is consistent with a low level of the toxic and feeding-deterrent sesquiterpene caulerpenyne, which is instead abundant in the congeneric inedible *Caulerpa taxifolia* [5]. On the other hand, *C. cylindracea* contains much higher levels of the harmless bisindole alkaloid caulerpin (CAU) [3], exhibiting a panel of biological activities of interest in food science and pharmacology [6,7], including its action as an agonist of peroxisome proliferator-activated receptors (PPARs) [3]. According to relevant literature [8,9,10,11,12], this latter finding suggests a possible involvement of CAU in the regulation of fish metabolism, food intake, and reproductive function. CAU has also been shown to accumulate in the liver, brain, and muscles of native fish feeding on the “exotic alga” [13,14], suggesting that it could also accumulate in fish gonads and offspring, with consequences on fertility and embryo development. To effectively explore the role of CAU in fish trophic and reproductive behavior under laboratory conditions, we employed the zebrafish (*Danio rerio*), a powerful experimental animal model in both genetic and developmental research, widely utilized for the study of fish nutrition, growth, and reproduction [15,16]. A sample of pure CAU isolated from *C. cylindracea*, whose ^1^H NMR spectrum is shown in Figure 1, was employed for all experiments.

## 2. Results and Discussion

### 2.1. Effects of CAU on Fish Voracity

Food with added CAU was prepared at a concentration of 0.1 percent CAU, approximately ten times higher than the concentration measured in *Caulerpa cylindracea*, according to a procedure we already employed for feeding experiments on fish [3]. The treated food was then offered to separate groups of adult zebrafish, evaluating the effects of CAU on food intake. Three separate trials were conducted on different groups of 10 fish each, in comparison with three different control groups. Trials were preceded by habituation phases, to familiarize fish with the test procedure, during which time the different groups showed a slightly different trophic activity, measured as the time in minutes needed to completely consume equivalent doses of the control food (Figure 2a,c,e).

In the trials, CAU-treated food was then administered to the groups who overall took longer to consume the offered food during the habituation phase (Figure 2**,** red line). During all trials, increased voracity in the fish fed with CAU-treated food was observed, with a progressive decrease in the time needed to completely consume the proposed food (Figure 2b,d,f). On average, fish took 12 min to completely consume the CAU-treated food in the first trial, whereas the control group took 30 min. These results were confirmed by the second and third trials, with an average of 13 and 11.5 min needed to completely consume the CAU-treated food, respectively, compared to an average of 32 and 30 min in the control groups. Differences in fish voracity were highly significant especially during the last feeding sessions (Figure 2, boxes on the right), suggesting that the greater palatability of the CAU-treated food could be associated with a progressive reinforcement of reward memory, possibly involving the endocannabinoid system, which is known to regulate food intake and reward memory in both fish and mammals [17,18,19]. This hypothesis is also consistent with the signaling crosstalk between the endocannabinoid system and PPARs [20,21], the main molecular targets of CAU [3], as well as with previous studies showing that PPARγ agonists cause hyperphagia and consequent weight gain in rodents [22].

### 2.2. Changes in the Expression Levels of Genes Controlling Food Intake and Metabolism

According to the behavioral data, the significant overexpression of the genes encoding PPARα and PPARγ was observed in the brain of adult fish treated with CAU, along with an upregulation of the genes *Cnr1* and *Cnr2*, encoding type 1 (CB1) and type 2 (CB2) cannabinoid receptors, respectively (Figure 3a). *Cnr1* was especially upregulated, supporting its major role in regulating eating behavior and energy balance related to food availability [23]. The existence of a cross-talk between orexinergic (hypocretinergic) and endocannabinoid systems has been reported in the literature, with both systems having common physiological functions in the regulation of appetite and reward [24]. In particular, an overlapping distribution of orexin and endocannabinoid receptors has been recently described in the brain of adult zebrafish [25]. Accordingly, along with the upregulation of *Cnr1* and *Cnr2* in the brain of CAU-fed fish, the gene *Hcrt*, encoding the peptide (pre-pro-orexin) from which both orexins A and B are derived, turned out to be significantly overexpressed (Figure 3a). The increased voracity after CAU administration was also associated with significant upregulation of the gene *5-HT1A* encoding for the serotonin 1A receptor in the fish brain (Figure 3a), supporting further cross-talk mechanisms between the brain endocannabinoid system and the serotonin system [26], with the participation of *5-HT1A* in stimulating food consumption in normally fed animals [27]. Overall, these results further confirm the existence of a complex interplay between the PPARs pathways and the endocannabinoid, orexinergic, and serotonergic systems in the stimulation of food intake. In parallel, the expression level of the gene encoding for the anorexigenic factor pro-opiomelanocortin (*Pomc*) was significantly downregulated in CAU-treated fish (Figure 3a). POMC is a precursor post-translationally processed that, in fish, generates melanocortin peptides (α and β-MSH), adrenocorticotropic hormone (ACTH), and other hormones [28]. It is worth mentioning that POMC-derived melanocortins are known to inhibit food intake in rodents and humans [29]. In fish, *Pomc* is mainly expressed in the pituitary gland and suppresses appetite by releasing α-MSH and inhibiting NPY neurons [30]. However, a strong downregulation of the gene encoding for the neuropeptide Y (*Npy*) was observed in the treated fish. Since NPY is known for its role in the stimulation of food intake, this finding seemingly contrasts with the increased fish voracity in fish treated with CAU. This inconsistency could be explained, on the one hand, by the increased expression in zebrafish of the gene encoding peptide YY (*Pyy-a*), which is known to inhibit Npy [31] (Figure 3a). On the other hand, the genes encoding the two isoforms of the hormone leptin (*Lep a* and *Lep b*), inhibiting Npy [32], also turned out to be overexpressed in CAU-fed zebrafish (Figure 3a). Leptin is a multifunctional hormone, involved not only in the control of food intake but also in reproduction and development processes [33]. A further inconsistency concerns the previous finding of an upregulation of Npy in *D. sargus* feeding on a CAU-enriched food [34]. In this case, however, the small sample size (*n* = 2) per fish group and the lack of information on fish voracity and/or leptin gene expression levels do not allow for effective comparisons. *Ppar**α* and *Ppar**γ* genes were also overexpressed in zebrafish livers, along with *Acadm*, the gene encoding for the acyl-coenzyme A dehydrogenase (Figure 3a, right), confirming the role of CAU as a mediator of lipid metabolism [3].

### 2.3. Effects of Dietary CAU on Lipid Metabolism

The changes in the expression levels of genes involved in the control of food intake were reflected by the increase in the absolute amount (on a dry weight basis) of fatty acids (FAs) in the white muscles of fish treated with CAU (Figure 3b), with a significant increase, in particular, in the levels of oleic acid (18:1 ω9), whose presence in the human diet appears to reduce the risk of coronary heart disease and to prevent diabetes mellitus [35,36]. This evidence is especially relevant when considering a possible use of CAU as a functional ingredient in aquaculture, which could help to improve the nutraceutical properties of fish flesh for human consumption. The absolute quantification of lipid levels was carried out by GC–MS by adding an IS, thus providing more accurate information than previous relative estimates based on ^1^H NMR spectroscopy [37].

### 2.4. Parental Transfer of Dietary CAU to Zebrafish Offspring

A preliminary evaluation was carried out on the white sea bream *Diplodus sargus*, a native fish of commercial interest, caught in a Mediterranean area characterized by high levels of *C. cylindracea* proliferation. This led us to detect relatively high levels of CAU in the fish gonads (Figure 4a) and to hypothesize the possible parental transfer of CAU to offspring, with consequences on fish fertility and embryo development. To test this hypothesis, we continued employing the zebrafish model, which can be easily reared under laboratory conditions. Food added with CAU (0.1 percent by weight) was administered to a group of ten adult zebrafish at a 1:1 sex ratio for ten weeks, and embryos obtained from the third to the tenth week were then analyzed for CAU quantification. This led us to measure relatively high amounts of CAU in the embryos at 32 h post-fertilization (hpf) obtained from all reproductive sessions, demonstrating the parental transfer of CAU to the fish eggs and developing offspring (Figure 4b). However, a time-dependent decline in the amount of accumulated CAU was observed in the developing larvae, with the total disappearance of the alkaloid in one-month-old larvae (Figure 4c).

### 2.5. Impact of CAU on Fish Reproductive Performance

To evaluate the effects of CAU on the offspring’s development and viability, fish groups at equal sex ratios were fed with CAU-treated food and compared with groups fed with control food. After three weeks of dietary treatment, two reproductive sessions at an interval of one week from each other were monitored during the first experiment (Figure 5a). This led us to observe a decreased rate of egg degeneration and a significant increase in both hatching rate and larval survival (viability) under treatment with CAU. The whole experiment was repeated once more on different groups of fish, and three further reproductive sessions were monitored (Figure 5b), confirming that the oral administration of CAU significantly improves zebrafish reproductive health and performance.

### 2.6. Effects of CAU on Zebrafish Gonadal Differentiation

Significant treatment–control differences in both gonadosomatic index (GSI), which is a good indicator of the reproductive activity, and hepatosomatic index (HSI), which reflects liver participation in vitellogenesis (Figure 6a,b), were also observed. After 21 days of treatment with CAU, GSI significantly increased in both males and females (Figure 6a), while HSI was lower in females only (Figure 6b). This concurrent increase of GSI and the decrease of HSI in female fish might be due to the transcription of sex-related genes during gonadal development, as previously observed in female rainbow trout [38]. Indeed, HSI increases at an early stage of ovary maturation in various fish species, while it decreases at a late stage of ovary maturation [39]. Accordingly, transversal histological sections of the ovaries of juvenile zebrafish deriving from the mating of fish treated with CAU for 21 days and fed with CAU-treated food for up to 120 days of development showed evident morphological differences relative to controls (Figure 6c). In all CAU-treated females (*n* = 6), a higher number of mid-, late-, and post-vitellogenic oocytes was observed, while the efferent ducts of the testes in CAU-treated males (*n* = 4) appeared to be larger and to have a more conspicuous mass of spermatozoa compared to the control fish (Figure 6c). Overall, the obtained results are consistent with the action of CAU as a dual PPAR α/γ agonist already reported in the literature [3]. PPARs are known to play critical roles in reproduction and embryonic development [40,41,42], sensibly affecting ovarian biology and trophoblast differentiation and maturation [11,12].

### 2.7. Changes in the Expression Levels of Reproductive Genes

The expression levels of genes involved in the control of the hypothalamic–pituitary–gonadal (HPG) axis were investigated after 3 weeks of dietary treatment with CAU. As expected, the significant upregulation of the genes encoding the two α and γ isoforms of PPARs was observed in the brains of both male and female fish (Figure 7a,b), while no significant differences in the expression of *Pparα* were detected in the gonads (Figure 7c,d). Instead, the gene encoding the PPARγ isoform was strongly downregulated in the ovary (Figure 7c) and upregulated in the testis (Figure 7d). The obtained data might be due to the advanced stages of oogenesis observed in CAU-treated fish (Figure 6c). Indeed, the expression of *Ppar**γ* is highly dynamic, increasing until the large follicle stage, followed by downregulation during terminal follicular growth, while ovarian expression of *Pparα* is relatively stable across the ovulatory cycle [43,44]. On the other hand, the upregulation of *Ppar**γ* in the testis is consistent with its role in the regulation of key lipid metabolic genes in Sertoli cells [45] and may reflect the appearance of larger efferent ducts in CAU-fed fish, with a more conspicuous mass of spermatozoa than in the testis of control fish (Figure 6c). A significant increase in the expression levels of the gene encoding the gonadotropin-releasing hormone (*Gnrh2*) was also observed in the brains of male and female fish, while a significant upregulation of follicle-stimulating hormone subunit beta (*Fshb*) and luteinizing hormone subunit beta (*Lhb*) genes only occurred in female brains (Figure 7a). An upregulation of gonadotropin genes, most likely associated with increased estrogen levels, might explain the early oocyte maturation observed in fish treated with CAU. This hypothesis is supported by the upregulation in the brains of CAU-fed fish of *Cyp19a1b*, which is the gene encoding for the type b aromatase involved in the final conversion of testosterone to estradiol (Figure 7a,b). On the other hand, the gene encoding the androgen receptor (*Ar*) turned out to be upregulated in both sexes under the effects of CAU, while the genes encoding estrogen receptors 1 and 2a (*Er1* and *Er2a*) were downregulated in female fish. In males, however, only the gene encoding for the ER1 isoform was significantly downregulated, possibly due to the inhibitory action of the aryl hydrocarbon receptor, whose gene (*Ahr1a*) was upregulated in the brains of CAU-fed fish of both sexes (Figure 7a,b), on estrogen receptors, in response to elevated circulating estrogen levels [46].

Sex-dependent differences in the expression pathway of kisspeptin genes (*Kiss1* and *Kiss2*) were also observed under treatment with CAU (Figure 7a,b). Although kisspeptins stimulate the release of GnRH, which in turn stimulates the production and release of gonadotropins, in teleosts, the argument appears to be a bit controversial; since studies in knock-out models have shown that kisspeptin is not essential for reproduction [47]. In the present work, instead, dietary treatment with CAU resulted in the strong downregulation of both kisspeptin isoforms in female brains, while these genes were significantly upregulated in males (Figure 7a,b). These findings were associated with improved reproductive performance and might be explained, on the one hand, by negative feedback exerted by circulating estrogens on *Kiss* genes [48], inhibiting *Kiss2* in the ovary (Figure 7c). On the other hand, *Kiss2* upregulation in the testes (Figure 7d) is consistent with a modulation of steroidogenesis and sperm function by kisspeptin signaling [49].

Overall, almost all investigated genes were upregulated in the brain and downregulated in the ovaries of fish treated with CAU (Figure 7). This could be due to negative feedback inhibiting downstream genes when the ovary is completely differentiated. It is well known that zebrafish are asynchronous spawners with follicles at different stages of development simultaneously present in the ovary [50]. Accordingly, the morphological analysis of the ovary carried out in the present study showed an early and more intense sexual maturation of the females treated with CAU compared to the control (Figure 6), consistent with a downregulation of *Ppar**γ* in the ovary (Figure 6c) [51]. The picture appears opposite in the testes, where most of the investigated genes turned out to be upregulated, except the genes encoding for gonadotropins (Figure 7d). The high expression levels of the vitellogenin 1 gene (*Vtg1*) were also observed in the livers of both male and female fish (Figure 7e). Since the expression of vitellogenin genes is regulated by estrogens [52], the obtained results support the action of CAU on estrogen modulation, resulting in the tissue-dependent regulation of reproductive genes.

## 3. Materials and Methods

### 3.1. Zebrafish Maintenance

Adult zebrafish (*Danio rerio*) used for the experiments were maintained at the facility for aquatic animal models at the Institute of Biomolecular Chemistry (ICB) of the National Research Council of Italy (CNR), approved by the Italian Ministry of Health (permission n. 20/2016-UT), according to the guiding principles provided by the ICB body in charge of animal welfare (*Organismo Preposto per il Benessere Animale—OPBA*). Fish were bred in the autonomous multi-tank system, ZebTEC Active Blue Stand Alone, produced by Tecniplast (Varese, Italy), which is a self-cleaning system that is equipped with automated a water-filtration and -purification system that allows maintaining the high levels of water quality necessary for a healthy aquatic environment. Adult zebrafish were held in 3.5 L tanks with pH values automatically kept between 6.8 and 7.5, temperatures between 26 and 28.5 °C, and a 14 h:10 h light/dark cycle. A daily change of about a third of the water, with the addition of water produced by a reverse-osmosis filtration system (Millipore RiOs 8 Essential) was made, while the salinity was kept under control by adding sea salt (Instant Ocean Sea Salt). A system of UV lamps allowed us to control the proliferation of pathogenic bacteria within the system. The fish were fed twice a day with granular dry food (SDS-400, Dietex, France). Embryos resulting from fish mating were rinsed into Petri dishes containing egg water (60 mg sea salt added to 1 L reverse-osmosis water) and incubated at 28 °C. Starting from day 6, free-swimming larvae were maintained on system water and fed twice daily with both paramecia and young fry diet (SDS-100, Dietex, France). Food size was then increased starting from day 20 (SDS-200, Dietex, France).

### 3.2. Isolation and Identification of CAU

*Caulerpa cylindracea* was collected in the Gulf of Pozzuoli (southern Italy) and stored at −20 °C until the chemical work. The extraction was performed with acetone, followed by partition between diethyl ether and water according to the literature [3]. Subsequently, the diethyl ether extract was concentrated and analyzed by silica gel thin-layer chromatography (TLC), using precoated Merck F254 plates, to assess the presence of a spot with R*f* around 0.35 in petroleum ether/diethyl ether (1:1), producing the red color typical of CAU after spraying with ceric sulfate. The diethyl ether extract was then divided into aliquots, each of which (1.0 g) was fractionated by chromatography on Sephadex LH-20 (Amersham Pharmacia Biotech; column diameter: 3 cm, h: 130 cm, LH-20: 200 g) using a solution of CHCl_3_/MeOH (1:1) in isocratic fashion to get 80 fractions that were combined on the basis of their TLC chromatographic behavior to give a main fraction containing CAU. This fraction was further purified by silica gel column chromatography using Merck Kieselgel 60 powder (70–230 mesh), eluting with an increasing polarity gradient of petroleum ether/Et_2_O, to give pure CAU (orange prisms) identified by the comparison of ^1^H-(Figure 1) and ^13^C-NMR-recorded data in DMSO-d6 with the literature values [53,54,55]. NMR spectra were recorded on a 400 MHz Bruker Avance III HD spectrometer equipped with a CryoProbe Prodigy.

### 3.3. Food Preparation

Treated and control food was prepared by modifying a procedure already described in the literature for feeding experiments on marine fish [3], according to the zebrafish dietary requirements. Control food was prepared by soaking dry granular food (SDS-100, SDS-200, SDS-400, distributed by Dietex, France) in acetone (1 mL/g dry food) and then evaporating the organic solvent under reduced pressure. Treated food was prepared in the same manner but after dissolving CAU (1 mg/g dry food) in the acetone before adding it to the dry food. This procedure led to a homogeneous distribution of CAU within the food.

### 3.4. Feeding Trials

Dietary treatments with CAU were conducted under a project approved by the Italian Ministry of Health (authorization No. 2/2019-PR of 03/01/2020). For the feeding trials, two tanks were set up, each containing ten six-month-old fish of an equal age and sex ratio. Before starting treatments, a habituation phase was carried out with the aim of allowing the fish to get used to the housing conditions and overcome the stress induced by being moved to the new tanks and estimating an adequate amount of food to be administered, not consumed too quickly, allowing the easy comparison of the time taken by the different groups of fish to fully eat the proposed food.

The habituation phase was carried out for one week, during which two groups of 10 fish (average weight 0.50 g) were fed with control food, administered in two daily doses of 50 mg each. This dose ensured that the fish took a relatively long time to completely consume the food (30–50 min). In the subsequent feeding trials, the time taken for the consumption of the control food was compared to the time taken to consume the food added with CAU at a dosing regimen of 10 μg/g body weight. The treatments were carried out for 21 days. The entire experiment, including the habituation phase, was repeated in triplicate on different groups of fish.

### 3.5. Fatty Acid Analysis

Total lipids were extracted from dried samples of zebrafish white muscle by the MTBE/MeOH protocol as previously described, by adding 20 µg of C23:0 methyl ester as an internal standard (IS). Bound FAs were converted into their corresponding methyl ester derivatives (FAME) by methanolysis. Lipid extracts were dissolved in 500 µL of methanol, and a tip of a spatula of sodium carbonate (Na_2_CO_3_) was added. The mixture was left to react overnight at 45 °C. The reaction mixture was then diluted with MilliQ water, and the basic solution was neutralized with a few drops of hydrochloric acid (HCl) [6 M]. The aqueous reaction mixture was subsequently extracted twice with diethyl ether (Et_2_O); the organic phase was transferred into pre-weighed vials and dried under an N_2_ stream and under vacuum to remove solvent traces. Free FAs were converted into their corresponding methyl esters (FAMEs) by adding an excess of an ethereal solution of diazomethane (CH_2_N_2_) freshly prepared in-house. After a reaction time of 60 min, samples were dried under an N_2_ stream, dissolved in dichloromethane (200 µL), and transferred into autosampler vials for FAME analysis by GC–MS. GC–MS analysis was carried out on an ion-trap mass spectrometer operating in EI mode (70 eV) (Thermo-Scientific, Polaris Q) connected with a gas chromatographic system (Thermo-Scientific, GCQ) equipped with a 5% phenyl/methyl polysiloxane column (30 m × 0.25 mm × 0.25 µm, Agilent, VF-5ms) using high-purity helium as the gas carrier. The following temperature gradient was applied: initial 160 °C holding for 3 min; then 5 °C/min up to 260 °C followed by 30 °C/min up to 310 °C, holding for 3 min at 310 °C; split flow 10 mL/min; full scan m/z 50–450. Injection of 2 µL. Analytical runs were processed by using Xcalibur software (vers. 3.0.63, Thermo Scientific, San Jose, CA, USA). FAME peaks were identified by comparing their elution times and MS spectra with a commercial standard pool (SIGMA-Aldrich, Marine PUFA-3, 1 mg mL^−1^) and the NIST database. For quantitative measurement, the peak area (A) of each FAME (x) was normalized by the peak area of the C23:0 ME IS and expressed as ng/mg dry weight (DW) as follows:FAME·(µg/mg·DW) = (A_x_ ∗ 20)/(A_IS_ ∗ mg·DW)

### 3.6. Quantification of CAU in Diplodus Sargus Gonads

Twelve individuals of *D. sargus* were sampled along the Apulian coast (southern Italy) in sites characterized by high levels of *C. cylindracea* proliferation. Gonads were immediately excised, frozen in liquid nitrogen, and maintained at −80 °C until chemical analyses, which were carried out according to Gorbi et al. (2014) [14]. After lyophilization, gonad samples were extracted with ethyl acetate by homogenization with a pestle and ultrasound. Indole-3-acrylic acid was converted into its methyl ester by reaction with diazomethane and added as internal standard (IS, 80 ng/mg, lyophilized tissue). After the removal of the organic solvent, extracts were reconstituted in MeOH at a final concentration of 0.5 mg/mL. Organic extracts were then analyzed by ultraperformance liquid chromatography/mass spectrometry (UPLC–MS/MS) to quantify the algal metabolite.

UPLC–MS/MS analyses were carried out on the Acquity UPLC system (Waters, Milford, MA, USA) online, with an API 3200 triple quadrupole mass spectrometer (ABSciex, Foster City, CA, USA) using multiple reaction monitoring (MRM) analysis. A Turbo VTM interface equipped with a turbo ion-spray probe in negative ionization mode was used. Data processing was performed on Analyst and Multiquant software packages (ABSciex). UPLC analyses were carried out on an Acquity CSH Fluoro Phenyl column (Waters, 100 × 2.1 mm, 1.7 µm,) maintained at 45 °C. The mobile phase consisted of a MeOH: H_2_O gradient from 40 to 95% of MeOH in 3.5 min at 0.45 mL/min. The injection volume was 2 µL. Three MRM transitions were monitored (397.0/365.0 *m/z*; 397.0/337.0 *m/z,* and 397.0/278.0 *m/z*) for the algal metabolite identification, and the MRM transition (200.2/168.2 *m/z*) was used for the IS. The quantitation of CAU was achieved by selecting the most intense transition at 397.0/365.0 *m/z*. The calibration curve was constructed by adding a known amount of CAU in MeOH and consisted of 2 blank samples and 6 calibration points (in triplicate) at concentrations in the range of 1 to 3000 ng/mL. The resulting peak areas under MRM trace were measured and plotted against concentration.

### 3.7. Reproductive Performance in Zebrafish under Treatment with CAU

Groups of fish (*D. rerio*) with equal sex ratio were mated once a week under dietary treatment with CAU. Spawning was stimulated in the morning when the light was turned on. Two hours later, the total number of laid eggs was recorded. Embryos were then examined under a stereomicroscope to calculate the fertilization rate. During the development of embryos/larvae, the number of degenerated embryos at 10 h post-fertilization (hpf), the number of hatched embryos at 72 hpf, and the number of viable larvae at 5 days post-fertilization (dpf) were recorded to compare the reproductive performance of fish treated with CAU with that of groups of fish fed control food.

### 3.8. Quantification of CAU in Embryos and Larvae

Pools of embryos (32 hpf, 50 embryos per pool), larvae (7 dpf, 30 larvae per pool), and juvenile fish (30 dpf, 30 fish per pool) were stored in Eppendorf tubes at −80 °C until extraction for the quantification of CAU. Before extraction, frozen embryos and larvae were freeze-dried. Subsequently, 10 μL of a solution containing indoleacrylic acid methyl ester (IS) at a concentration of 100 μg/mL and 240 μL of methanol were added to each tube. The samples were then homogenized by vortexing and methyl tert-butyl ether (MTBE) was added (750 μL). After sonication, the samples were stirred for one hour at r.t. MilliQ water (187.5 μL) was added to each sample to induce phase separation and was then centrifuged at 4 °C at 1000 g/minute for 10 min. The organic phase was recovered in pre-weighed vials. The extraction was repeated with 250 μL of MTBE, the organic phases were recombined, dried under an N₂ stream and vacuum, weighted, and stored at −20 °C until analysis. The quantitative determination of CAU was carried out by UHPLC-HRESIMS by internal standard calibration. The chromatographic method [56] relied on a biphenyl column (Phenomenex, 150 × 2.1 mm, 1.7 µm) and an elution gradient of the two solvents A (H_2_O) and B (MeOH) programmed as follows: from 40 to 80% of B in 2 min, then the percentage of B increased to 100% in 13 min, remaining there for 7 min. Flow: 0.3 mL/min. T column: 28 °C; injection volume: 5 µL. The mass spectrometry method used a high-resolution spectrometer Q-Exactive (Thermo Scientific), a Quadrupole-Orbitrap hybrid system with an electrospray source (HESI) in negative polarity. The instrumental parameters were as follows: spray voltage −3.0 kV, capillary temperature 320 °C, S-lens RF level 55, auxiliary gas temperature 350 °C, sheath gas flow rate 60, auxiliary gas flow rate 35. To build up the calibration curve, 5 concentration levels of CAU standard were used: 3, 10, 30, 100, and 300 ng/mL, with IS at 5 µg/mL. The construction of the calibration curve was carried out considering the ratio between the area corresponding to the molecular ion [M-H]- of CAU (extracted ion at 397.1194 *m/z*) and that of the IS (extracted ion at 200.0717 *m/z*) as a response, with respect to the concentrations used of CAU (y = 0.001014 + 0.000433552x, R^2^ = 0.9907). For LC–MS analysis, the extracts were re-dissolved in 200 µL of MeOH and transferred into autosampler vials.

### 3.9. Assessment of Gonadosomatic (GSI) and Hepatosomatic (HSI) Indexes

Following euthanasia, the body weight of each fish was measured using an electronic balance, along with the weights of gonads and livers. This allowed us to calculate the gonadosomatic index (GSI) and the hepatosomatic index (HSI). In particular, the GSI was calculated from the ratio of total body weight and gonad weight as follows:

GSI = wet gonad weight (mg) × 100/wet body weight (mg).

The HSI was calculated, instead, from the ratio of total body weight and liver weight:

HSI = wet liver weight (mg) × 100/wet body weight (mg).

The brains, gonads, and livers of each fish were then immediately stored at −80 °C for subsequent RNA extraction.

### 3.10. Histological Analysis of Gonads in Juvenile Zebrafish

A pool of 20 embryos resulting from the mating of both experimental groups was grown to evaluate the possible effects of CAU on gonadal development. Embryos born from CAU-fed fish were in turn fed with CAU-treated food (SDS-100, Dietex, France) starting from 5 dpf, while embryos resulting from the control group were fed with the same amount of control food. Treatments were carried out until 120 dpf, and food size was increased according to the stage of development. After euthanasia, the heads and bodies of juvenile zebrafish were fixed in Bouin’s solution (5% acetic acid, 9% formaldehyde, 0.9% picric acid) for 24 h, dehydrated in graded ethanol, and embedded in paraffin wax [57]. Approximately 5–7 mm thin sections of gonads were transversally cut and stained with hematoxylin and eosin.

### 3.11. RNA Extraction and Analysis

Zebrafish brains, gonads, and livers were homogenized in 0.5 mL TriReagent (Sigma-Aldrich, Germany). RNA isolation was performed according to the manufacturer’s protocol. Total RNA was quantified using NanoDrop ND-1000 Spectrophotometer (Thermo Scientific), and RNA samples with a 260/280 nm ratio between 1.9 and 2.1 were selected for further analysis. cDNA was obtained from 1 μg RNA/sample, using a Quantitec Reverse Transcription Kit (Qiagen, Hilden, Germany), following the manufacturer’s instructions. The mRNA expression level of the different zebrafish genes was analyzed by RT-qPCR, using the iQ5 Real-Time PCR Detection System (Bio-Rad, Hercules, CA, USA). PCR reactions were carried out in a total volume of 15 μL using 10 ng of cDNA, 4 pmol of each primer, and 7.5 μL 2× Luna Universal qPCR Master Mix (New England BioLabs, Ipswich, MA, USA). Cycling parameters were as follows: one cycle at 95 °C for 3 min and 40 two-step cycles at 95 °C for 10 s and at 60 °C for 30 s. Samples were run in triplicate, using 96-well qPCR microplates (Bio-Rad). All quantifications were normalized to the expression of 18S rRNA endogenous control (*Rps-18*). Values of cycle threshold (Ct) obtained in quantification were used for the calculations of fold changes in mRNA abundance according to the 2^−ΔΔCt^ method [58]. PCR primers (Table 1) were designed by using the open-source *Primer3* software (https://primer3.ut.ee/ accessed on 3 May 2020).

## 4. Conclusions

The present study highlights the potential of CAU, a metabolite produced by *Caulerpa cylindracea*, which is one of the most threatening invasive algae in the Mediterranean Sea, to modify food web structure and dynamics, by affecting fish trophic behavior and reproduction. The oral administration of purified CAU to zebrafish resulted in an evident increase in fish voracity, with a highly significant decrease in the time needed to completely consume the proposed food, compared to fish fed with a control diet. This evidence strongly supports that CAU is responsible for the altered behavior of Mediterranean fish, which have changed their eating habits by including as a main component of their diet an invasive alga that is particularly rich in this bioactive alkaloid [1,2]. On the other hand, dietary CAU significantly ameliorated the whole reproductive process in the zebrafish model. This suggests that the observed negative effects of a *C. cylindracea*-based diet on the health of Mediterranean fish [1,13,14,59] are not due to the action of CAU but are perhaps due to other algal metabolites (e.g., the toxic sesquiterpene caulerpenyne) and/or to a relative nutritional depletion in the diet of omnivore fish that have switched to predominantly herbivorous habits. Instead, the functional properties of CAU highlighted in the present report, which seem to be an expression of its direct interaction with PPARs [3,10,11,12], may offer a suitable strategy for the valorization of the huge algal biomass produced by *C. cylindracea* along Mediterranean coasts. Such “undesired” biomass could be harvested and processed to obtain CAU: a functional ingredient of interest for the development of new diets for fish farming, which could reduce feed wastage by increasing fish voracity, at the same time enhancing fish reproductive health. However, the present study also paves the way for possible nutraceutical applications of CAU in the veterinary and biomedical fields, towards the development of new functional foods that can help face problems related to the reduction of appetite and reproductive efficiency.

## Figures and Tables

**Figure 1 marinedrugs-20-00513-f001:**
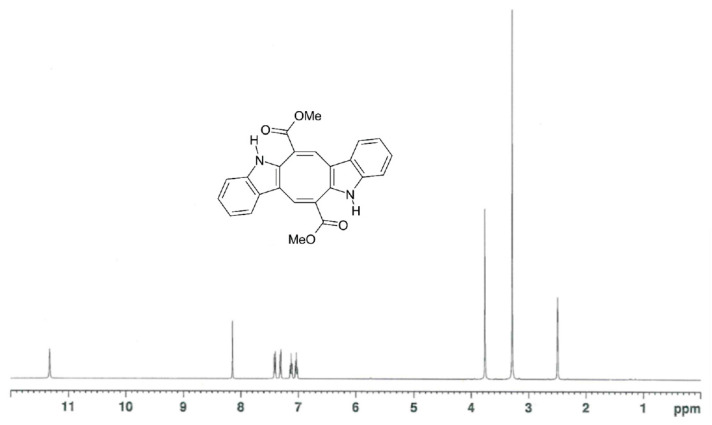
^1^H NMR spectrum of caulerpin (CAU) isolated from *C. cylindracea* (400 MHz, DMSO-d6).

**Figure 2 marinedrugs-20-00513-f002:**
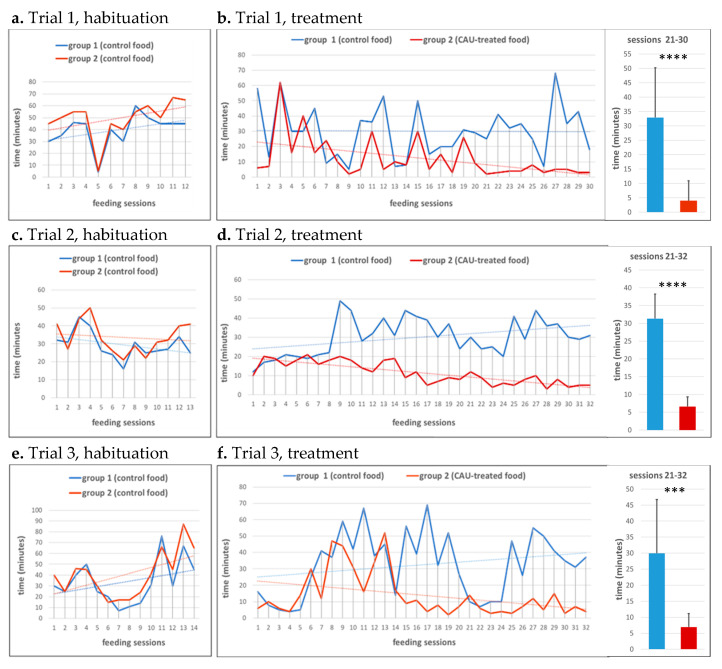
Time spent for food consumption by different fish groups during three separate feeding trials. Food was administered in two daily doses, one in the morning (odd numbers) and one in the afternoon (even numbers). Each trial was preceded by a habituation phase (**a**,**c**,**e**) in which fish were fed with control food. In (**b**,**d**,**f**), food added with CAU was administered to the group indicated with a red line, while the group represented with a blue line continued to receive control food. In the boxes on the right, the average times employed by the two different groups to completely consume the proposed food, during the last 10–12 sessions, are shown as mean ± standard deviation. Significant differences between the means of the two groups were evaluated by using the unpaired two-tailed *t*-test (α = 0.05; *** *p* < 0.001, **** *p* < 0.0001).

**Figure 3 marinedrugs-20-00513-f003:**
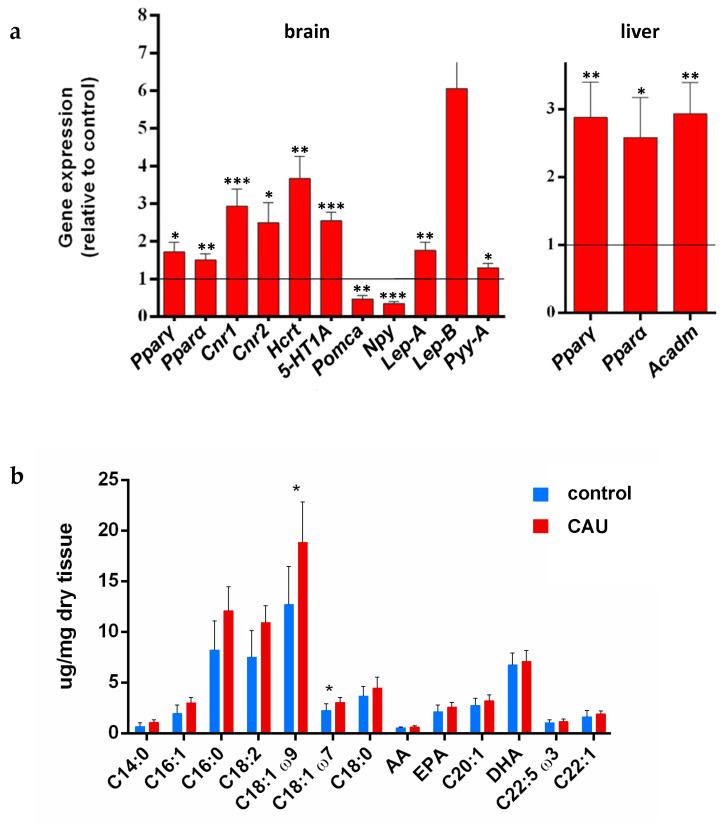
Relative changes in gene expression and lipid profiles. (**a**) (left) mRNA levels of genes involved in the central control of food intake in the brain of adult zebrafish after 21 days of treatment with CAU-enriched food and (right) relative mRNA levels of genes involved in the lipid metabolism in the fish liver. Values were normalized to the values of the control, set at 1 as indicated by the line. Data are shown as mean ± standard deviation (*n* = 7 for each bar and the control; α = 0.05; * *p* < 0.05; ** *p* < 0.01; *** *p* < 0.001). (**b**) Levels of total fatty acids (measured as methyl ester derivatives) in the white muscle of fish treated with CAU compared with fish fed with control food. Data are shown as mean ± standard deviation (unpaired *t*-test, *n* = 7 for each bar; α = 0.05; * *p* < 0.05).

**Figure 4 marinedrugs-20-00513-f004:**
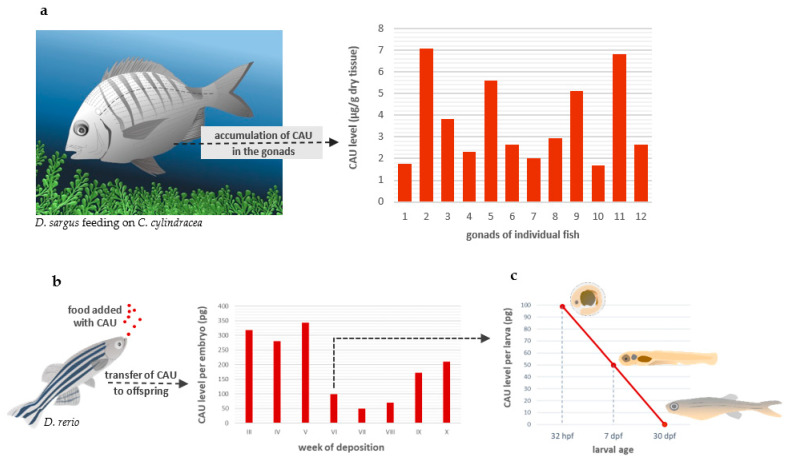
Bioaccumulation of CAU. (**a**) CAU accumulates in the gonads of wild-caught *D. sargus* feeding on *C. cylindracea* (*n* = 12). (**b**) Parental transfer of dietary CAU to zebrafish offspring during eight subsequent reproductive sessions. (**c**) Time-dependent decline in the amount of CAU accumulated in eggs laid at week VI. Hpf: hours post-fertilization; dpf: days post-fertilization.

**Figure 5 marinedrugs-20-00513-f005:**
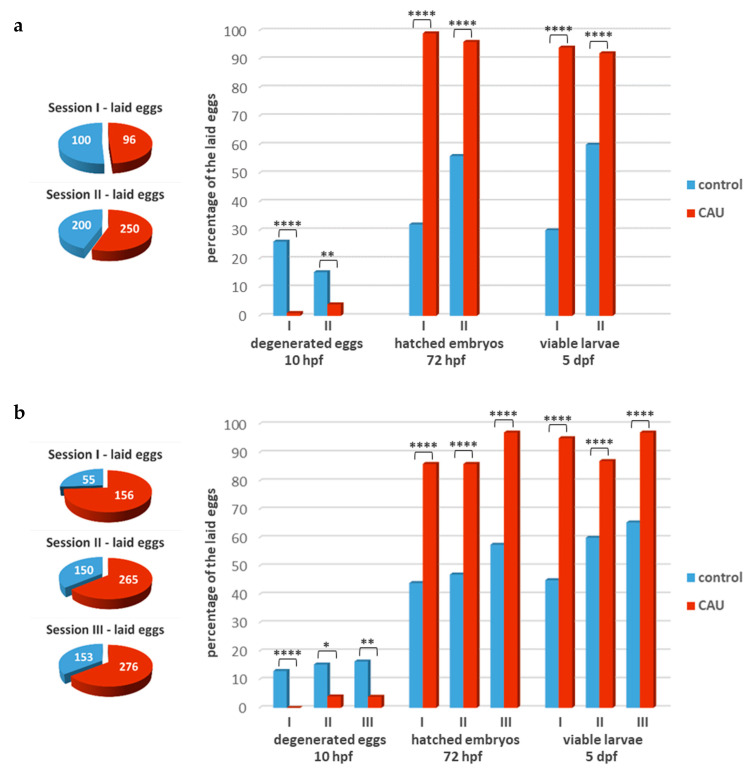
Effects of CAU on fish reproduction. (**a**) Reproductive performance, in terms of the percentage of degenerated eggs, hatched embryos, and viable larvae, in fish treated with food supplemented with CAU. Performance was monitored during two different reproductive sessions one week apart (I–II), in comparison with control groups; (**b**) the above experiment was repeated on different groups of fish during three subsequent reproductive sessions (I–III). Significant differences were evaluated by using the two-tailed χ^2^ test (α = 0.05; * *p* ≤ 0.05, ** *p* ≤ 0.01, **** *p* ≤ 0.0001).

**Figure 6 marinedrugs-20-00513-f006:**
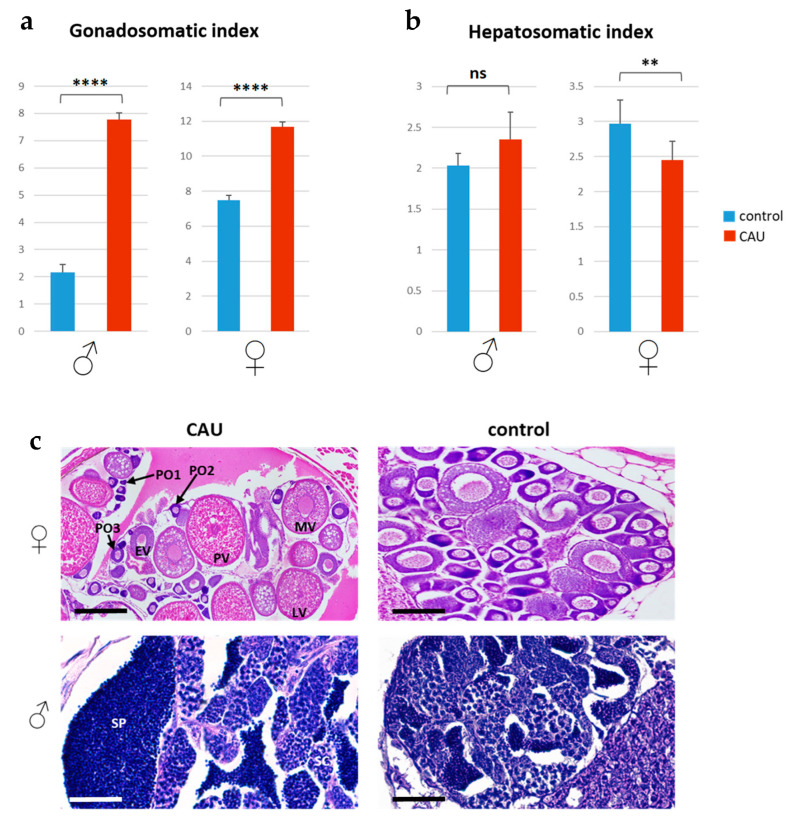
Morphometric analyses. Effects of dietary CAU on the gonadosomatic (**a**) and hepatosomatic (**b**) indices after 21 days of treatment in male (*n* = 4) and female (*n* = 6) adult zebrafish; results are represented as the mean ± standard deviation (significant differences were evaluated by using the unpaired *t*-test, α = 0.05; ** *p* < 0.01, **** *p* < 0.0001); (**c**) histological sections of ovaries and testes (stained with H-E) of CAU-treated and control fish; PO1, primary oocytes, stage 1; PO2, primary oocytes, stage 2; PO3, primary oocytes, stage 3; EV, early-vitellogenic oocytes; MV, mid-vitellogenic oocytes; LV, late-vitellogenic oocytes; PV, post-vitellogenic oocytes; SP, spermatozoa; SC, spermatocysts. Scale bar: 200 µm.

**Figure 7 marinedrugs-20-00513-f007:**
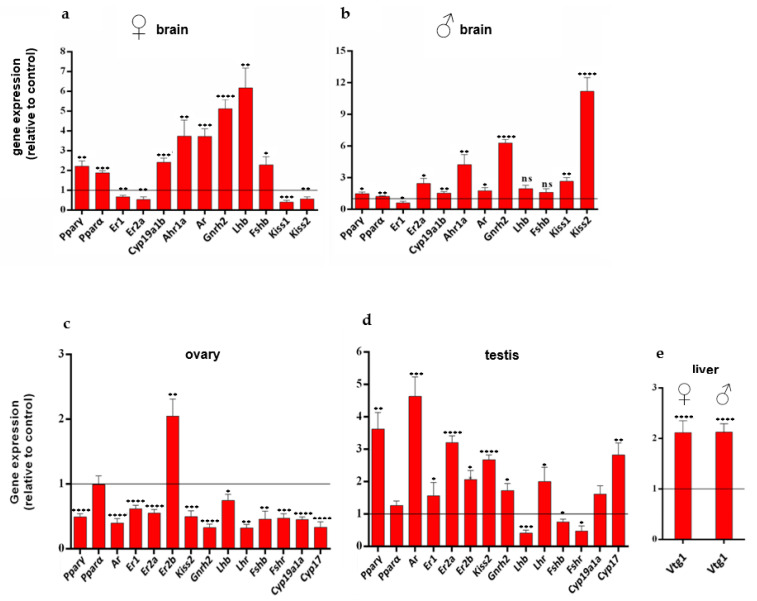
Relative mRNA levels of genes involved in the control of the HPG axis. Relative changes in gene expression in (**a**) female brains (*n* = 7), (**b**) male brains (*n* = 7), (**c**) ovaries (*n* = 6), (**d**) testes (*n* = 4), and (**e**) livers (*n* = 7) in zebrafish after 21 days of treatment with CAU-enriched food. Values were normalized to the values of the control, set at 1 as indicated by the line. Data are shown as mean ± standard deviation (unpaired *t*-test, α = 0.05; * *p* < 0.05; ** *p* < 0.01; *** *p* < 0.001; **** *p* < 0.0001).

**Table 1 marinedrugs-20-00513-t001:** List of qPCR primers.

*Name*	*Forward*	*Reverse*
***Er*1**	CACAGGACAAGAGGAAGAAG	ATGGTGATCTCTGTGTAGGG
***Er*2*a***	TCCGAAAGTGCTATGAAGTT	TTATCTCTTGAGACCTCGGA
***Er*2*b***	AAAGCCATACACTGAGGCTA	CAGATCTCCACATCAATCCT
** *Ar* **	CAAAGCCGTGTCCGTATC	TTCGCCTCTGTCTCGTCCC
***Cyp*19*a*1*a***	CGCAGAGAAACTTGACCATTC	CGCATCACCATCTCCAACAC
** *Fshr* **	GGCAACACCGAAGACACAC	CGTGTAGTTCAGACAGGGCT
***Cyp*19*a*1*b***	CAGTCGTTACTTCCAGCCATTC	CCGCTGTTTCTCCGTTGC
***Cyp*17**	TCTTTGACCCAGGACGCTTT	CCGACGGGCAGCACAA
** *Lhr* **	CCTGGAGGCTCATTTCAT	GAGATTCATTGTGGCGTAT
***Ahr*1*a***	ATGACATGAATGGTGTTGGAGAG	ACTGTTCCGATGTAAGCTTGT
**5-*HT1A***	AATCATCGGCTCGCTTTTCC	TAAGGTCTGTAACGGCCAGG
*C* ***nr*1**	TACTGGAAGAGGTCAATC	AGAGTCAATAGTGAGCAA
*C* ***nr*2**	ATTGCAAGCTCCACAGCACT	AAACGCCATTGTGACGCCA
***Ppar*-*α***	TCCACATGAACAAAGCCAAA	AGCGTACTGGCAGAAAAGGA
***Ppar*-*γ***	CTGCCGCATACACAAGAAGA	TCACGTCACTGGAGAACTCG
** *Fshb* **	ATGAGGATGCGTGTGCTTGTTC	GTGATGGAGATGTTGGTGAGTCG
** *Lhb* **	GGCTGGAAATGGTGTCTT	GGCTCTTGTAAACGGGAT
** *Lep b* **	TTCCCCGTCACCTCCAACTA	CCTTGCATGTGCCATTGTGTT
** *Lep a* **	TTCCCCGTCACCTCCAACTA	CCTTGCATGTGCCATTGTGTT
** *Acadm* **	AAGGTTTTGAGGGCAGGTGT	ACTCTTTCTGCTGCTCGGTT
***Kiss*1**	CAAGCTCCATACCTGCAAGTG	GTACCCTCGCCACTGACAAC
***Kiss*2**	CAGAGCCTATGCCAGACC	CTAGTCGATGTTTGCAGGATATTT
** *Hcrt* **	AGTGCATCTCAACAACGACG	GTGAGTTGTGCAGCAGTTGT
** *Npy* **	ACAAAGCCCGACAACCCG	AGCGCTTGACCTTTTCCCAT
***Pyy*-*a***	CGCGCTGAGACACTACATCA	GTGCTCTGTGTCATCCCCAA
***Gnrh*2**	CCAGGACTGCAGTAGAGGAG	GCACTCAGACACAGCATCAG
***Vtg*1**	GTCGCTGTTCCCATCAATCC	GCAGTACAGCAGTGGTCTAA
** *Pomca* **	CTGTCGAGACCTCAGCACAG	GCTTTCTCCAGGGTAGACGG
***Rps*-18**	TGGTGTGGCTATGAACCCTG	TGGACGGTCTTTGTTCCTCG

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
