# Peer review of "An Alkaloid from a Highly Invasive Seaweed Increases the Voracity and Reproductive Output of a Model Fish Species"

_marinedrugs, 2022, doi:10.3390/md20080513_

Round 1
Reviewer 1 Report
This study is based on the previous finding (by several of the same authors) that the alkaloid caulerpine is present in a highly invasive Mediterranean algae that became a preferred dietary item for several autochthonous fish species. The characteristics of caulerpine as a regulator of several metabolic pathways and its effect on feeding rate and reproduction in fish, following previous work, led the authors to test the hypothesis that this compound is involved in the feeding preference of fish for the invasive algae, which they tested by simple experiments using treatments with caulerpine-bearing food and their respective controls. For the analysis of the differences between the treated and control groups the authors use a variety of techniques that together elegantly inform on the processes behind the behavioral and physiological differences resulting from the experiments, opening up a range of new avenues of research into the metabolic processes involved, their ecological effects and some potential practical uses in fish culture and more.
Overall the manuscript is well written, the methodologies are clear and sound, and the results of the various analyses are coherent and converge into solid conclusions. I enjoyed reading this work and believe that it will be of interest to readers more familiar with the specific topics covered by Marine Drugs but also for those more interested in the ecological aspects of introductions of exotic species (and their "new" compounds).
I have made some comments, questions and minor suggestions for you to consider, that you will find attached to the reviewed version of your MS.

Author Response
We are deeply grateful to Reviewer #1 for his/her appreciation of our work and for the important help in improving the manuscript. The manuscript has been revised in accordance with his/her suggestions. Below are our responses to all comments.
- Comment 1 (title). This title is a bit misleading since suggest that you found effect on "fishes". Though the generalization does not sound unlikely, you actually proved it for just one fish species. I suggest using a more accurate title such as "An alkaloid from a highly invasive seaweed increases the voracity and reproductive output of a fish" or something in the like.
Answer. The title has been changed to “An alkaloid from a highly invasive seaweed increases voracity and reproductive output of a model fish species”
- Comment 2 (line 142). Why have you chosen to show the results as standard deviations in the first case and as standard errors in the second? In my opinion, only confidence limits are visually useful, although I accept that they have the unpleasant tendency to be too large to look good in figures. Whatever your preference, I suggest you to consider keeping only one measure of variation along the text.
Answer. We are sorry, there was a mistake in writing the legend to figure 3. Indeed, in both cases the standard deviation was displayed. This error has been corrected in the revised version.
- Comment 3 (line 160). It was not obvious to me that this was also your own work and is part of the original herein presented results until I read the M&M section below. Maybe you can explicit that here.
Answer. The following passage has been added at the beginning of section 2.4 (lines 190-194): “A preliminary evaluation was carried out on the white sea bream Diplodus sargus, a native fish of commercial interest, caught in a Mediterranean area characterized by high levels of C. cylindracea proliferation. This led us to detect relatively high levels of CAU in the fish gonads (Figure 4a), and to hypothesize a possible parental transfer of CAU to offspring, with consequences on fish fertility, and embryo development.”
- Comment 4 (figure 5). This effect is a very interesting finding of yours and likely has a link with the enhanced feeding presented above. Also I guess you noticed that there seems to be a trend of diminishing the relative value of the effect of CAU treatment through spawns... Have you any idea to explain that (discarding it is not a artifact of some sort of course).
Answer. Unfortunately, we cannot currently offer a valid explanation for this trend, but we thank the reviewer for this observation that paves the way for further research.
- Comment 5 (line 298, line 325). Was this protocol also maintained during the experiments? If so, what are the chances that in cases of treatment with diets containing CAU, a fraction of the compound added in food will dilute and accumulate in the water over time? I think it should not be a problem but would be interested in knowing if I understood it right and if you considered it. Could that alter the results significantly? If so, How?
Answer. Caulerpin is insoluble in water. In addition, food residues are filtered by the system before the water returns to the breeding tanks. However, even assuming that extremely low traces of the compound could have reached the control fish through the medium, they evidently did not prevent to detect highly significant differences between the effects of cau-treated vs control food.
- Comment 6 (line 361). In here and through the whole manuscript: To me it is a good practice writing the full species name each time it is mentioned for first time in a section.. otherwise one has to scroll up and down to remember which genus/species is. Particularly in cases where to genuses start with the same letter.
Answer. The full species name has been now mentioned in each section when it is mentioned for the first time.
- Comment 7 (line 437). I guess that this is wet weight, not dry weight, but it should be stated here.
Answer. Yes, it is the wet weight, as now indicated in the revised version.
- Comment 8 (line 513). Is cylindracea also competitively displacing other autochthonous food items, mostly other edible algae with similar palatability features?
Answer. Although the C. cylindracea invasion is negatively affecting many keystone Mediterranean species, its caulerpin-mediated ability to increase the voracity of fish, described in this work, has not yet been highlighted for any native alga. Although Mediterranean fish such as D. sargus are known to also eat small amounts of other algae, including the autoctonous Caulerpa prolifera, this behavior is not comparable with the observed marked preference for Caulerpa cylindracea. This is in agreement with our results, as C. prolifera does not contain caulerpin, while it does contain much higher levels of the toxic sesquiterpene caulerpenyne.
Reviewer 2 Report
the author provide the evidence of caulerpin to effect the food intake, gene, reproductive performance and possible mechanism to model fish Danio rerio, and caulerpin was a major chemical constituent of Caulerpa cylindracea. the manuscript was important to protect environmental and ecology. However, aouthor should add the content of caulerpin in Caulerpa cylindracea. Author always used the pure caulerpin to test the bioactivity, whether the whole algae Caulerpa cylindracea influence the above-mentioned indicators? how many it will effect? whether it accumulate in fish?
Author should the NMR data and purityof caulerpin. whether there are any lipid in the Caulerpa cylindracea?because so many chemical constitutents is from the Caulerpa cylindracea and fish eat the whole agale.
Author Response
We are very grateful to Reviewer #2 for highlighting the importance of the ecological/environmental aspects of our work. We also thank the reviever for raising critical questions regarding the purity of the sample of caulerpin we employed for the assays, the level of caulerpin in Caulerpa cylindracea, and the possible effects of other algal metabolites on fish. Answering to these questions allowed us to significantly improve manuscript readability.
- We answered the first question by adding the 1H-NMR spectrum of the purified caulerpin in figure 1.
- To answer the second question, we clarified (at the beginning of section 2.1) that the chosen concentration of caulerpin was “approximately ten times higher than that the concentration measured in cylindracea, according to a procedure we already employed for feeding experiments on fish [3]”
- About the third question, it is worth emphasizing that although other algal metabolites could produce significant and different effects on fish behavior and physiology, our study only focused on caulerpin because of the molecular interactions of this compound with molecular targets relevant for the regulation of metabolism and reproduction (see Marine Drugs 2018, 16, 431). This allowed us to candidate purified caulerpin as a possible food additive of interest for possible applications in aquaculture and/or in the veterinary and biomedical field. However, we agree with Reviewer #2 that a seaweed-based diet could produce very different effects than simply administering purified caulerpin.
Finally, as suggested by Reviewer #2 , the English style has been revised
Reviewer 3 Report
Schiano et al studied the effects of caulerpin, from an invasive seaweed Caulerpa cylindracea, on the voracity and reproductive performance of model fish Danio rerio. The manuscript exhibits interesting experimental results related to caulerpin. The manuscript has sufficient novelty to be published in Marine Drugs. I recommend publication of this manuscript. However, following minor corrections should be revised before publication.
- The title of the manuscript should be revised. My recommendation is "Effects of caulerpin on the on the voracity and reproductive performance of the model fish Danio rerio"
- Abstract, Line 25, "secondary metabolite" can be used instead of "major chemical constituent".
- Line 37. This reference should be cited: "Paul, N.A.; Neveux, N.; Magnusson, M.; de Nys, R. Comparative production and nutritional value of “sea grapes”—The tropical green seaweeds Caulerpa lentillifera and C. racemosa. J. Appl. Phycol. 2013, 26, 1833–1844."
- Line 52-54. Figure 1 is not necessary, could be removed.
-Page 3, the figure 2c. "contro" should be "control" and why there is a sharp decline in the 5th feeding session?
- Line 77, there is no red line in Figure 1. it must be Figure 2.
- Line 159-160, Diplodus sargus and C.cylindracea should be written in italic.
- Line 308, Purity of CAU should be given. was it %100?
- Line 312, Even if a reference is given, a short detail should be given for the isolation of CAU such as column diameter, length, mobil phases etc.
- Line 325, if possible, a video link can be provided related to feeding of fishes,
-Line 500, the Mediterranean "Sea"
- A General Question: What are the differences in general blood parameters (such as ALT, AST) in control and experimental groups?
- Line 536, Caulerpa cylindracea should be written in italic
-Line 542, Caulerpa taxifolia var distichophylla should be written in italic
-Line 559, Caulerpa racemosa should be written in italic
-Line 564, Danio rerio should be wirtten in italic
- Line 610, Diplodus sargus should be written in italic
- Line 613, Oncorhynchus mykiss should be written in italic
- Line 640-641, 644, 653 Caulerpa spp should be written in italic
Author Response
We would like to thank Reviewer #2 for his/her valuable comments on the manuscript. Below are our responses to all comments.
- Comment 1 (title). The title of the manuscript should be revised. My recommendation is "Effects of caulerpin on the on the voracity and reproductive performance of the model fish Danio rerio".
Answer: Combining the suggestions by both Reviewer #1 and Reviewer #3, the title has been changed to “An alkaloid from a highly invasive seaweed increases voracity and reproductive output of a model fish species”.
- Comment 2 (line 25). "Secondary metabolite" can be used instead of "major chemical constituent".
Answer: "Major chemical constituent" has been changed to "major secondary metabolite".
- Comment 3 (line 37). This reference should be cited: "Paul, N.A.; Neveux, N.; Magnusson, M.; de Nys, R. Comparative production and nutritional value of “sea grapes”—The tropical green seaweeds Caulerpa lentilliferaand racemosa. J. Appl. Phycol. 2013, 26, 1833–1844."
Answer: The article by Paul et al. (2013) has been cited. Many thanks for this suggestion.
- Comment 4 (line 52-54). Figure 1 is not necessary, could be removed.
Answer: According to the comment by Reviewer #2, the 1H-NMR spectrum of the purified caulerpin has been added in figure 1, to account of the purity of the assayed compound. We therefore considered it appropriate to still show the structure of caulerpin to facilitate the reading of the NMR spectrum. However, if Reviewer #3 deems it strictly necessary, we can remove it.
- Comment 5 (page 3). the figure 2c. "contro" should be "control" and why there is a sharp decline in the 5th feeding session?
Answer: "Contro" has been changed to "control". We too were surprised by the sharp decline in the 5th feeding session (trial 2). At the moment, we do not know how to explain this, since all feeding sessions were conducted under controlled conditions of temperature, light, conductivity and pH. Perhaps some event took place in the facility (which was not recorded), influencing the feeding behavior of the fish. On the other hand, the similar trends of the curves in the habituation phases suggest that the two groups of fish, placed in adjacent tanks, influenced each other. In this case, more prominence should be attributed to the differences observed under the action of CAU during the trials.
- Comment 6 (line 77). There is no red line in Figure 1. it must be Figure 2.
Answer: “Figure 1” has been changed to “Figure 2”.
- Comment 7 (line 159-160). Diplodus sargus and C.cylindracea should be written in italic.
Answer: Done.
- Comment 8 (line 308). Purity of CAU should be given. was it %100?
Answer: The 1H-NMR spectrum of the purified CAU used for the experiments, now shown in figure 1, testifies to a purity very close to 100%.
- Comment 9 (line 312). Even if a reference is given, a short detail should be given for the isolation of CAU such as column diameter, length, mobil phases etc.
Answer: More details have been provided in the M&M (lines 342-352).
- Comment 10 (line 325). if possible, a video link can be provided related to feeding of fishes.
Answer: During some preliminary tests we tried to record videos, but we noticed that the presence of the operator considerably disturbed the behavior of the fish. Maybe we should have automated the shooting, but we didn't.
- Comment 11 (line 500). the Mediterranean "Sea"
Answer: “Mediterranean” has been changed to “Mediterranean Sea”
- Comment 12 (a general question). What are the differences in general blood parameters (such as ALT, AST) in control and experimental groups?
Answer: We did not evaluate differences in general blood parameters.
- Comment 13 (line 536). Caulerpa cylindracea should be written in italic.
Answer: Done
- Comment 14 (line 542). Caulerpa taxifolia var distichophylla should be written in italic.
Answer: Done
- Comment 15 (line 559). Caulerpa racemosa should be written in italic.
Answer: Done
- Comment 16 (line 564). Danio rerio should be wirtten in italic.
Answer: Done
- Comment 17 (line 610). Diplodus sargus should be written in italic.
Answer: Done
- Comment 18 (line 613). Oncorhynchus mykiss should be written in italic
Answer: Done
- Comment 19 (lines 640-641, 644, 653). Caulerpa spp should be written in Italic.
- Answer: Done